# No Pain, No Gain? In Defence of Genetically Disenhancing (Most) Research Animals

**DOI:** 10.3390/ani9040154

**Published:** 2019-04-09

**Authors:** Katrien Devolder, Matthias Eggel

**Affiliations:** 1Oxford Uehiro Centre for Practical Ethics, University of Oxford, Oxford OX1 1PT, UK; 2Institute for Biomedical Ethics and History of Medicine, University of Zurich, 8006 Zurich, Switzerland

**Keywords:** bioethics, animal ethics, genetic disenhancement, animal research ethics

## Abstract

**Simple Summary:**

Millions of animals are used for scientific purposes in the EU every year. The procedures they undergo often cause significant pain, suffering and distress. New gene editing technologies now potentially offer a new and feasible way to genetically modify research animals in order to reduce or eliminate their ability to feel pain and to suffer. In this paper, we discuss the ethical concerns this new technology and new possibility raise and evaluate the implications of such genetic modifications with regards to the legal regulations in animal research in Europe.

**Abstract:**

Every year, around 12 million animals are used for the purpose of scientific research in the European Union alone. The procedures performed on them often cause significant pain and suffering. Despite regulations aimed at reducing this suffering, we can expect millions of research animals to continue to suffer in the near to mid-term future. Given this reality, we propose the use of gene editing to create research animals with a reduced capacity for suffering, in particular, from pain. We argue that our proposal would be in line with moral principles embedded in European regulations regarding animal research, and that it would facilitate compliance with these regulations. We also respond to the strongest argument against our proposal—the ‘no pain no gain’ argument.

## 1. The Suffering of Research Animals

Every year, around 12 million animals are used in scientific research in the European Union (EU) alone [1]. The procedures performed on them often cause significant pain and suffering [2]. There is a broad consensus that sentient animals have a moral status, and thus, that there are moral constraints on what we can permissibly do to them in order to advance our own goals, including the goal of protecting human health [3]. These moral constraints are reflected in European legislation regulating the use of animals in scientific research. Directive 2010/63/EU [4] (henceforth, the Directive) requires that such research complies with the 3Rs (Replace, Reduce and Refine), a set of principles originally developed by Russell and Burch in the late fifties [5,6]. *Replace* encourages researchers to replace animal experiments with non-animal models, such as, for example, computer simulations; *Reduce* encourages them to use the fewest animals possible to achieve results which are consistent with the study’s objectives; and *Refine* to refine research procedures and the way animals are cared for in the laboratory so as to minimise suffering [6]. The Directive also instructs scientists to demonstrate, in a harm-benefit-analysis, that “the harm to the animals in terms of suffering, pain and distress is justified by the expected outcome taking into account ethical considerations, and may ultimately benefit human beings, animals or the environment” [4], article 38.2.(d)). These two requirements set out in the Directive—compliance with the 3Rs and demonstration in a harm-benefit analysis that the expected benefits outweigh the expected harms—capture the idea that preventable harm to research animals is ethically problematic, and that suffering inflicted on them is only legally and morally justified if it is outweighed by the experiments’ expected benefits to humans, animals or the environment. 

Although the Directive’s ultimate goal is to stop all animal experimentation ([4], preamble 10) it is unlikely that the use of animals for research purposes, and thus, research animals’ suffering, will end any time soon. In the EU, the number of animals used for scientific purposes has remained more or less constant over the last 10 years [1,7], and there is currently no indication that this number will substantially decrease in the near or mid-term future. 

## 2. Animal Disenhancement

We propose tackling the problem of research animals’ continued suffering by using gene editing to create disenhanced research animals with a reduced capacity for suffering, in particular from pain. Paul Thompson coined the term ‘animal disenhancement’ to refer to the altering of animals to better suit their environment [8]. In what he calls the ‘dumb down’ approach, one begins with a complete animal and removes through, for example, genetic engineering or selective breeding, certain capacities or functions. Animal disenhancement has been the topic of heated debate in the context of the consumption of meat, dairy and eggs [8,9,10,11,12,13,14,15]. Our proposal in this paper is inspired by Adam Shriver’s recent defence of animal disenhancement to reduce animal suffering in the context of factory farming [13,16]. Shriver argues that we should replace all animals in factory farms with animals that have been genetically engineered to lack enzymes in the brain which are responsible for enabling the affective dimension of chronic or persistent pain. The animals would still be able to feel and react to the stimulus that would normally be painful, but would suffer less from it since it would no longer be perceived as painful; it would no longer cause a negative subjective experience (we will return to this in more detail) In this paper, we focus on genetic disenhancement to reduce the suffering of research animals by intervening in the affective dimension of pain. However, there are also interesting ethical issues raised by the possibility to reduce animal suffering by creating research animals that lack consciousness. On some views of harm, this would be more acceptable than our proposal, since it would be the moral equivalent of not bringing an animal into existence. Reducing suffering from pain is, according to some, less acceptable because it makes the animal worse off than it would otherwise have been. One of the authors (name removed for review process) addresses this issue in a different paper (in progress). 

In this paper, we propose the application of a similar approach to research animals. We argue that creating research animals with a reduced capacity to suffer from pain would (i) in many cases be in line with the widely accepted moral obligation to reduce suffering if we can (3Rs), (ii) facilitate the harm-benefit analyses in project evaluations required by the Directive, and (iii) challenge, and reduce the scope of, the ‘no pain no gain’ argument against the widespread use of pain relief (including via pharmacological means) in research animals. 

Though more research into using gene editing to create animals with a reduced capacity for suffering from pain is required to ensure that this does not interfere (or at least, not too much) with the validity of the research, we think that once this practical hurdle is out of the way, the reasons for applying our proposal of animal research are, in many cases, stronger than those against it.

Note that we do not address here the issue of whether animal research or the genetic modification of animals in general are ethically acceptable; we are only dealing with the specific question of the moral permissibility or desirability of creating genetically-disenhanced research animals to reduce research animals’ suffering. Animal experimentation and genetic modification of animals raise ethical issues regarding respect for the intrinsic value and Telos of animals. However, these issues have been discussed in depth elsewhere in the literature [3,16,17]. These papers have convincingly argued that objections to disenhancing or genetically modifying animals appealing to disrespect of Telos or intrinsic value do not stand up to ethical scrutiny. We did not want to reiterate these arguments but decided to focus on issues that were particular to the debate about genetically disenhancing research animals. We think that gene editing should be used to reduce various instances of suffering, but will focus on suffering from pain as an example of one possible approach for how to do this. For the purpose of this paper, we are also accepting the 3Rs (as moral and legal principles) and the Directive without criticism. 

## 3. Genetic Disenhancement with CRISPR Gene Editing 

With the advent of CRISPR (Clustered Regularly Interspaced Short Palindromic Repeats), gene editing [18] has achieved unprecedented levels of precision, affordability and simplicity. Concerns have been raised that CRISPR might lead to unintended changes to DNA, or so-called off-target effects. However, it is still unknown how severe the risks for such off-target mutations are. Thus, it has been stressed that it is important to investigate the frequency and risks for such off-target effects, especially with regards to the application of CRISPR in the clinic [19,20]. It has been claimed that, given recent advances in our understanding of the molecular processes regulating pain, it could soon become technically feasible to use gene editing to disenhance animals in ways that reduce their capacity to suffer from pain without also causing major side-effects [13,16]. We restrict ourselves here to briefly recapitulating recent scientific developments regarding genetic modifications of pain in animals. (For more detail, see [13,16]).

Pain research has shown that pain can be divided into two distinct dimensions that correlate to activity in different brain regions [21,22,23,24]. The sensory dimension of pain is associated with the primary and somatosensory cortex and constitutes the quality of the pain (e.g., a dull, pulsing, or burning sensation), its localization and intensity. The affective dimension of pain is associated with the anterior cingulate cortex and the insula cortex, and several studies indicate that it is relevant for what is believed to constitute the suffering aspect of pain, that is, how much one minds the pain or how unpleasant the pain is [21,25,26]. Importantly, it has been demonstrated that one can genetically modify the affective dimension of pain independently of the sensory dimension [23], resulting in patients who can still feel pain but do not mind it as much, or do not find it unpleasant, which greatly reduces their suffering [13]. This is similar to a patient who has been given morphine, who can still feel the pain but does not mind it as much, and thus, suffers less.

In his paper defending the creation of genetically-disenhanced farm animals, Shriver specifically refers to promising research involving the genetic modification of either cAMP (cyclic adenosine monophosphate, an important molecule for intracellular signalling [27] or peptide P311 (a short chain of amino acids highly expressed in the anterior cingulate cortex) in rodents in order to reduce their capacity to suffer from pain [13]. Interfering with the cAMP or P311 pathways could reduce the affective dimension of pain, that is, the ‘caring about’ the painful sensation [13] and/or chronic pain symptoms, while leaving the acute pain response intact [27]. Shriver suggests that either approach could potentially also be used to produce cows and pigs with a reduced capacity for suffering from pain. Though more research is needed to avoid unwanted side effects, such as impairment of vital physiological functions, it has been shown that cAMP is regulated at the local level [28] (i.e., regulation is spatially restricted to particular locations), which suggests its selective targeting may indeed be feasible. Recent findings regarding the function of two genes (FAAH and FAAH-OUT) suggest that it might not only be feasible to selectively target pain perception but also suggest that this might be possible without negatively affecting the ability to have positive experiences [29]. 

## 4. The Moral and Legal Imperative to Comply with the 3Rs

As mentioned earlier, one central aspect of the Directive is compliance with the 3Rs [5], which have also been widely accepted as an ethical framework for animal research [6]. The principle Refinement demands that the suffering of research animals is minimised, and their welfare improved whenever possible. The creation of genetically-disenhanced research animals with a reduced capacity for suffering would arguably be in line with Refinement, which not only provides a legal reason in favour of it, but also a moral reason, since the idea that we should prevent suffering whenever we can at a relatively little cost to ourselves is a widely accepted one. 

The Directive is concerned with harm to animals in terms of pain, suffering and distress (article 38 d). Pain is understood as an unpleasant subjective sensation. Suffering is understood as a long-lasting or permanent negative, subjective experience such as continuing pain, fear, or thirst. Genetic disenhancement holds the potential to greatly reduce pain, and suffering from pain. However, it is unclear what effect genetic disenhancement could have on fear and thirst. It is, for example, possible, that genetically-modified rodents would still suffer during the experiment because they are not accustomed to human handling or if they are not given enough water. Harm can also include the impairment of species-specific properties, functions and habits, such as species-specific social behaviour. Since genetically-disenhanced animals would still be sentient, that is, they would still be able to have positive experiences and sensations, and since there is no obvious reason to assume that genetic disenhancement would change the species-specific nature of the animals, it is likely that they should still be able to perform most of their species-specific behaviour. Based on this, it seems that genetic disenhancement could reduce harm and would be in line with Refinement.

However, some may deny that the ablation of the affective pain dimension would be compatible such Refinement by arguing that this is likely to result in an increase in injuries, such as bruising, and this may adversely affect the animal’s wellbeing. However, as mentioned earlier, studies have shown that eliminating the affective pain dimension can leave intact acute pain features and acute responses to noxious stimulation [13]. Thus, animals are likely to still exhibit normal guarding behaviour (i.e., behaviour aimed at avoiding pain), making an increase in self-injuries unlikely. Moreover, if the bruises are a result of abnormal guarding behaviour due to a lack of pain sensation, then this indicates that the bruises themselves would not cause any suffering. 

Injuries may, however, also occur because of an increase in careless treatment of animals that have a reduced capacity for suffering, and this may result in more overall suffering compared to the status quo. That this effect would occur is a claim that is difficult to prove or disprove, but note that an increase in careless interactions with research animals would be inconsistent with an existing moral and legal commitment to reduce animal suffering in scientific research, and with the acknowledgement of the intrinsic value of animals in current EU legislation [4], recital 12. However, we agree, that to be in line with *Refinement*, genetically-modified research animals must not be worse off than their non-gene edited variants, not only regarding their well-being as a consequence of their altered genetic make-up, but also regarding the way they are handled by researchers. If there is reason to believe they would be worse off than their non-gene edited variants, then this provides a strong argument against pursuing our proposal.

Others might concede that creating research animals with a reduced capacity for suffering is consistent with Refine, but nevertheless object to it on the grounds that it is inconsistent with the principles Reduce and Replace, and therefore, that it is not sufficiently in line with the Directive. They might argue that creating genetically-disenhanced animals would only treat the symptoms but not the cause of what is the real underlying problem of animal research: our general disrespect for animals. It provides a technological fix that ignores this problem, and, even worse, may even help to maintain it. After all, it could be that the creation of research animals with a reduced capacity for suffering legitimises the view that we can use animals as we please, and this could slow down the transition to morally preferable alternatives. Thus, some may argue that, rather than treating the symptoms through a technological fix, we should only (and immediately) address the real problem and cease all animal experimentation. We should *Reduce* and *Replace*, not *Refine*.

First, we also believe that we ought to eventually cease all animal experimentation if morally preferable alternatives are available. However, we take as our point of departure the reality that for the foreseeable future, large numbers of animals will continue to be used for research purposes, despite a moral and legal commitment in Europe (and the US) to phase out animal research. Until better alternatives are implemented, using genetically-disenhanced research animals could provide a viable solution to reducing animal suffering. Our approach is a harm reduction approach which is comparable to giving methadone treatment to drug addicts: ideally, we’d prevent drug addiction, but given that drug addiction can be expected to continue to exist, it is morally preferable to reduce harm to the addicts and others (e.g., those involved in drug crimes) by providing drug addicts with methadone, while also simultaneously fighting drug addiction.

Secondly, whether our proposal may in fact slow down the transition to morally preferable alternatives is a question that is difficult to answer and which will, among other things, depend on the extent to which we simultaneously support these alternatives. But even if we suppose, for the sake of argument, that our proposal can indeed be expected to slow down the transition to morally preferable solutions, does this provide a decisive reason against pursuing it? This seems implausible. Along the same reasoning we could argue that it would have been wrong to improve the welfare of slaves in the United States of the 18th and 19th century as this may have delayed the abolition of slavery. Even if it did eventually delay that, we think it was justified, and even morally obligatory to improve the slaves’ lives whenever the improvement was significant and there was no clear evidence that it would significantly delay abolition. Utilitarians may disagree, but even they may think that we should have done all we could to increase the welfare of slaves at the time when the risk of this causing a delay in the abolition of slavery was small, or uncertain. Since it is uncertain whether improving the welfare of research animals through genetic disenhancement will effectively delay the transition to the abolition of animal experiments, we think we should do it. 

Note that the ‘technological fixes argument’ has also been adduced against proposals to genetically disenhance animals in the context of agriculture. However, unlike in animal research, no strong commitment to moving to morally preferable solutions exist in the agricultural sector. For example, there is no intention embedded in legislation to phase out factory farms. This suggests that the risk of contributing to more animal experimentation through our proposal is significantly lower than the risk of contributing to more factory farms in the case of Shriver’s proposal [13,16].

Finally, *Replacement* is sometimes also understood as ‘relative replacement’ [30], that is, animals are not replaced per se, but cognitively ‘higher developed’ animals are replaced by ‘cognitively lower’ developed animals. The rationale is that the former are believed to have a higher capacity to suffer than the latter. On this understanding of *Replacement*, one could argue that cognitively disenhancing research animals to make them suffer less is an instance of relative replacement, and that it is thus in line with *Replacement* in the Directive.

In conclusion, we think that the creation of genetically-disenhanced research animals with a reduced capacity for suffering from pain is consistent with the 3Rs.

## 5. Facilitating Harm-Benefit Analysis in Project Evaluation

The EU Directive regulating animal research instructs scientists to demonstrate, in a harm-benefit-analysis, that the animal research they wish to perform is justified. In such project evaluations, the expected harms to the animals in terms of suffering and the expected benefits to humans, animals and the environment have to be identified and balanced against each other [4]. For a project to be approved, the benefits need to outweigh the harms. However, unlike defining harm, defining benefit at a conceptual level has proven difficult [31,32]. In addition, there is a practical problem regarding how to weigh the expected benefits of an experiment against expected harms. Practical benefits (e.g., the development of a new line of drugs) are generally very difficult to predict due to (i) the fact that for any scientific experiment, it is unpredictable whether the hypothesis will be verified (resolving this uncertainty is, after all, the whole point of conducting the experiment), and (ii) difficulties with translating knowledge obtained through animal models to human applications. This has led some to question about whether an estimation of the practical benefits of an experiment in project evaluation is even possible [31,32]. Perhaps the most we can hope for is an estimate of the *expected* knowledge gain of a particular experiment. But knowledge gain has typically been considered of lower significance than practical benefit [33,34,35,36,37], and consequently, it has been argued that it is far more difficult for expected knowledge gain (as opposed to practical benefits) to outweigh expected harm to research animals. Finally, there is the more general question of whether harm to animals can be weighed against benefits to humans at all; some have compared this process to weighing apples and oranges [38].

Creating genetically-disenhanced research animals with a reduced capacity for suffering could facilitate harm-benefit analyses in project evaluations because far less harm would need to be outweighed by the expected benefits. If the expected harm is significantly reduced, it becomes less problematic to be able to provide a definition of harm that is widely agreed upon. It would also make it easier for knowledge gain to be sufficient to swing the analysis to a positive verdict. Thus, not only would our proposal prevent difficulties associated with justifying animal suffering in the context of animal research, it would also help scientists meet the legal requirements of a positive harm-benefit-analysis in project evaluation.

## 6. The ‘No Pain, No Gain’ Argument 

So far, we have provided reasons for accepting our proposal to create genetically-disenhanced research animals with a reduced capacity for suffering from pain and have failed to find a sufficient reason to reject it. In the remainder of the paper, we address what we believe to be the strongest potential objection to our proposal, which appeals to what we call the ‘no pain, no gain’ argument. The basic idea is that for many experiments, reducing or eliminating research animals’ pain would negatively affect the validity of the experiment, and hence, should be avoided. It is an argument that has been adduced against the use of pain relief via pharmacological means in research animals [2]. It could also be adduced against our proposal. 

Though the 3Rs and the requirement of a positive harm-benefit analysis in the Directive provide a legal and moral framework for the avoidance or minimization of harm and suffering inflicted on research animals, they do not imply that causing or allowing pain in research animals is categorically prohibited. It can be morally and legally justified to do so if it is required to protect the validity of the experiment ([4], article 14). For example, in the study of pain killers, pain itself is the studied parameter. The study objective, therefore, requires that the control cohort is left untreated while the other cohort is treated with the pain killer. However, pain inflicted on research animals may also be a side effect and researchers are then faced with a choice whether to provide pain relief or not. Take, for example, researchers growing cancers in animals to study cancer treatments. The pain these cancers cause can be difficult to treat with painkillers; successful pain treatment may require continuous medication e.g., with strong opioids [2]. Not only is this type of pain management often considered too demanding a strategy in certain animals, such as rodents, but it is also known that opioids (such as morphine) and nonsteroidal anti-inflammatory drugs (such as ibuprofen) have unwanted side-effects throughout the body and may influence tumour progressions. Using these drugs may thus impair study validity. In a case like this, researchers may legally opt for leaving the pain untreated if doing otherwise could be expected to affect study results. The underlying idea is that in such cases, without pain, there is no gain. The same argument might potentially be used against genetic disenhancement of research animals. The possibility of negatively affecting study validity could serve as a strong argument against our proposal too. 

We agree that study validity is the cornerstone of justified animal research. We also acknowledge that genetic disenhancement is not well enough understood to exclude the possibility that it will negatively affect study validity. However, we do not take this as a conclusive objection to genetic disenhancement in research animals, but as a reason for conducting more research into potential unwanted effects of genetic disenhancement on study validity, and, importantly, for engaging in an ethical discussion about the implications of any such effects. As Larry Carbone pointed out [2], what we here refer to as the ‘no pain, no gain’ argument has been taken too much for granted and applied too widely, that is, to more cases of animal research than was (probably) necessary, at a significant cost to animals who have suffered more than they should have. There are several reasons why this has been the case, but the most relevant one to our paper is that it has often been too quickly assumed that pain relief is an unacceptable potential disruptor of study validity, while other potentially disrupting factors have been ignored or accepted. What other factors do we have in mind? First, there is pain itself. It is often forgotten or ignored in animal research that not only pain relief, but also experiencing pain itself, can have unwanted side-effects that threaten the validity of study results. Pain-induced perturbations of homeostatic mechanisms could, for example, induce stress, which also influences the physiology of an animal, and can thus also compromise study results [39]. An induced surgical myocardial infarction to study ‘heart attack’ in a mouse could trigger a painful inflammatory response and make breathing more difficult thereby reducing ventilation [2], and pain may make animals less likely to eat and drink. All these direct and indirect effects of experiencing pain could potentially affect study results [2]. Secondly, there are factors such as staff turnover, heterogeneous equipment, housing, breeding, and handling of animals, different protocols, different animal genetics and the availability of various medications that are all known to be potential sources of variability in the experiment [2]. But unlike pain relief, these factors and pain itself are generally accepted as potential disruptors of study validity.

Our proposal to genetically disenhance research animals highlights the importance of (re)opening the ethical discussion on the scope of the ‘no pain no gain’ argument. We need to decide how much study perturbation by genetic disenhancement (and other means of pain relief) is acceptable, and to what extent can the beneficial effects on animals’ welfare outweigh the costs of this study perturbation. After all, this is an open question, and perhaps we should accept more study perturbation if this could increase animal welfare. 

Our proposal not only re-opens the debate about the ‘no pain no gain’ argument, it may also reduce its scope. Consider an experiment in which the omission of pharmacological pain relief is, correctly, grounded in the ‘no pain, no gain’ argument. It is possible that in such a case, pain reduction via genetic disenhancement would have fewer unwanted side effects on study results than pharmacological pain relief. It could also be that, compared to doing the research on animals without any pain relief, using disenhanced animals increases the quality of the research, not only by reducing pain, but also by reducing stress experienced as a result of the pain, which can distort research results [8,15]. Thus, in such a scenario, the use of genetically-disenhanced research animals would reduce the number of cases the ‘no pain no gain’ argument would normally apply to, and thus, reduce its scope. The degree to which scientific validity is reduced by pain, pain alleviation (through pharmacological or genetic means), or other factors will differ significantly from case to case. That is why we think that using genetically-disenhanced research animals could be acceptable in most, but not all cases of animal research. Its acceptability will thus have to be decided on a case-by-case basis.

## 7. Conclusions

We propose the use of gene editing to create disenhanced research animals with a reduced capacity to suffer from pain. We have shown that our proposal is in line with moral principles embedded in European legislation regulating animal research. It is consistent with the principles *Refine* and *Replace*, and not necessarily inconsistent with the principle *Reduce*, as it would reduce suffering in research animals without necessarily leading to an increase in the numbers of animals used. Though we think that it would be preferable to completely replace animal use in research instead of refining it, we assume from a reality in which this will not happen any time soon. We have argued that refining animal use is preferable to the status quo. We take the concern that a technological fix will slow down the transition to morally preferable alternatives to animal research seriously, but do not think that, at present, there is sufficient reason to believe this will occur.

We have also shown that our proposal would facilitate harm-benefit analyses in project evaluations, a corner stone of the Directive. Our proposal would reduce the amount of harm that needs to be outweighed by knowledge gain, thus facilitating the weighing process and the justification of animal use in research. 

Animal research is only justifiable if it generates valid data. We have pointed out that, in the case of genetic disenhancement of research animals, we are dealing with a conflict of trying to minimise harm to the animals and optimizing research outcomes. Genetic disenhancement could lead to less suffering from pain and distress in research animals, thus potentially increasing study validity. At the same time, genetic disenhancement could also have detrimental unknown and unwanted side effects that could jeopardise scientific validity. This potential risk is, we think, the strongest reason to approach our proposal with caution. It does, however, not provide the basis for a conclusive objection to it, but rather a starting point for further research and discussion. More research must be done into the potential unwanted side effects of genetic disenhancement on study validity, and important ethical questions need to be addressed about what sort of study perturbations we are prepared to accept, to what extent, and at what cost to animal welfare. We thus advocate a re-evaluation of the ‘no pain, no gain’ argument and expect that this re-evaluation would lead to a narrowing of the scope of the argument. The scope of the argument could also be reduced if replacing research animals with genetically-disenhanced animals results in fewer cases where the argument could normally be applied.

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
