# Peer review of "No Pain, No Gain? In Defence of Genetically Disenhancing (Most) Research Animals"

_animals, 2019, doi:10.3390/ani9040154_

Round 1
Reviewer 1 Report
The authors introduce the concept of animal disenhancement to animals used for scientific purposes more in particular with respect to the suffering from pain. Their justification concentrates around the legal obligation to comply with the 3Rs and the 'no pain, no gain' argument. The authors conclude that disenhanced research animals with a reduced capacity for suffering from pain are in line with the concepts of the 3Rs and facilitate the harm benefit analysis. They advocate a reevaluation of the 'no pain, no gain' argument.
This reviewer regards the manuscript as an attempt to stimulate discussion among the scientific community using animals for scientific purposes. Unfortunately, the authors defer all the important issues for further discussion without addressing them more in depth first to provide a firm basis for these discussions.
With this in mind, the concept of intrinsic value needs to be addressed at least in the context of the principle of creating disenhanced animals. Then there is the notion that the disenhanced animals would still be able to feel pain, but would suffer less. This leaves the authors with the obligation to discuss the 'concept' of suffering beyond the technical explanation how pain and suffering can be attributed to different areas of the brain. With regard the latter, the authors are careful in not stating that this has been unequivocally established yet, which makes it even more necessary to address the different concepts of pain and suffering to a certain extent.
In conclusion this reviewer is of the opinion that certain central concepts of the manuscript require deepening before this manuscript can truly serve as "a starting point for further research and discussion".
Author Response
Review 1:
Black: reviewer 1
Red: Authors’ response
The authors introduce the concept of animal disenhancement to animals used for scientific purposes more in particular with respect to the suffering from pain.
Their justification concentrates around the legal obligation to comply with the 3Rs and the 'no pain, no gain' argument.
The authors conclude that disenhanced research animals with a reduced capacity for suffering from pain are in line with the concepts of the 3Rs and facilitate the harm benefit analysis.
They advocate a reevaluation of the 'no pain, no gain' argument.
This reviewer regards the manuscript as an attempt to stimulate discussion among the scientific community using animals for scientific purposes. Unfortunately, the authors defer all the important issues for further discussion without addressing them more in depth first to provide a firm basis for these discussions.
With this in mind, the concept of intrinsic value needs to be addressed at least in the context of the principle of creating disenhanced animals.
First of all, we would like to thank the anonymous reviewer for their constructive feedback.
Though the issue of intrinsic value of research animals is an important one, the reason why we don’t address it in this paper is that we are assuming, for the sake of argument, that animal experimentation and gene editing are ethically acceptable, and want to focus on arguments that haven’t yet been addressed in the debate about genetic disenhancement and animal experimentations. If one accepts animal experimentation then one accepts that animals can be used merely as a means to others’ ends, and that their intrinsic value is not necessarily respected. Thus, gene editing to make them suffer less when they are instrumentally used in any case does not result in less disrespect for their intrinsic value (and arguably could result in more respect for their intrinsic value). We have now clarified in the text (please, see footnote on page 2, line 80) why we do not address the issue of intrinsic value in this paper, and have referred to papers in which this issue has already been addressed in depth.
Then there is the notion that the disenhanced animals would still be able to feel pain, but would suffer less. This leaves the authors with the obligation to discuss the 'concept' of suffering beyond the technical explanation how pain and suffering can be attributed to different areas of the brain.
We agree that our formulation was ambiguous. We have now changed the relevant sentence on p. 2 lines 63-65 to:
„The animals would still be able to feel and react to the stimulus that would normally be painful but would suffer less from it since it would no longer be perceived as painful; it would no longer cause a negative subjective experience“
This should help clarify that genetically disenhanced animals would still be able to feel and react to external stimuli, but would not suffer as a result of them.
We have also added a paragraph in which we explain how the concept of harm (pain, suffering and distress) is defined in the EU Directive (please see page 3 and 4 lines 123-135) .
With regard the latter, the authors are careful in not stating that this has been unequivocally established yet, which makes it even more necessary to address the different concepts of pain and suffering to a certain extent.
In conclusion this reviewer is of the opinion that certain central concepts of the manuscript require deepening before this manuscript can truly serve as "a starting point for further research and discussion".

Reviewer 2 Report
I found this a really interesting and thought-provoking piece. I assumed it was very hypothetical as from my knowledge of CRISPR I suspect we are a long way away from doing this. I also suspect it would be very difficult to uncouple the affective pain response without influencing other body systems. Nevertheless it is certainly an interesting ethical debate. I wonder why you specifically focus on pain and not other negative experiences or even consciousness itself?
The essay obviously included your personal opinions against the various ethical theories widely-used in research. I was surprised that you did not mention objections based on intrinsic value or telos, or the slippery slope argument often cited when genetic modfication is discussed. I'm not suggesting you should include these but it might tick of the common ethical theories/arguments. I commend you in tackling this novel, and no doubt controversial topic.
My only specific query is below:
Lines 215-218- I'm not sure the objection against pain relief in research animals is due to pain reduction invalidating the experiment per se. I believe it is more usually that the pain management used may impact on some experimental outcomes eg inflammatory markers or healing therefore making it difficult to interpret the experiment. I also disagree with the statement discussing a lack of use of pain management (l218). I'm not sure whether there is up-to-date research on this, but I suspect pain relief is probably the norm rather than the exception where pain is significant.
Author Response
Review 2:
Black: Reviewer 2
Red: Authors’ response
Reviewer 2:
I found this a really interesting and thought-provoking piece.
I assumed it was very hypothetical as from my knowledge of CRISPR I suspect we are a long way away from doing this. I also suspect it would be very difficult to uncouple the affective pain response without influencing other body systems. Nevertheless it is certainly an interesting ethical debate.
I wonder why you specifically focus on pain and not other negative experiences or even consciousness itself?
First, we would like to thank the anonymous reviewer for the positive and constructive feedback. We chose to focus on pain as a study case, i.e. as one possible approach to reducing the suffering of research animals. There are indeed interesting ethical issues raised by using gene editing (or other means) to create animals with a reduced or no consciousness. Indeed, one of the authors is in the process of writing a different paper that specificially focusses on this issue.
We have now stated more explicitly that we are interested in the use of gene editing to reduce research animals’ suffering in general but that we focus on the reduction of pain as a study case, to illustrate how this could be approached (please, see p.1 line 21 – 22 and p. 2 lines 64-65). We also now acknowledge in a footnote that interesting issues are raised by comparing the ethics of using gene editing to reduce pain, and to reduce consciousness, and refer to the aforementioned work in progress. (please see page 2, line 55)
In addition, we have added a new paragraph in which we clarify that the EU Directive understands harm as pain, suffering and distress. We have included this to elaborate on different forms of potential negative experiences. (see page 3 and 4 lines 123-135)
The essay obviously included your personal opinions against the various ethical theories widely-used in research. I was surprised that you did not mention objections based on intrinsic value or telos, or the slippery slope argument often cited when genetic modfication is discussed. I'm not suggesting you should include these but it might tick of the common ethical theories/arguments. I commend you in tackling this novel, and no doubt controversial topic.
Though the issue of intrinsic value of research animals is an important one, the reason why we don’t address it in this paper is that we are assuming, for the sake of argument, that animal experimentation and gene editing are ethically acceptable, and want to focus on arguments that haven’t yet been addressed in the debate about genetic disenhancement and animal experimentations. If one accepts animal experimentation then one accepts that animals can be used merely as a means to others’ ends, and that their intrinsic value is not necessarily respected. Thus, gene editing to make them suffer less when they are instrumentally used in any case does not result in less disrespect for their intrinsic value (and arguably could result in more respect for their intrinsic value). We have now clarified in the text (please see footnote on page 2, line 80) why we do not address the issue of intrinsic value in this paper, and have referred to papers in which this issue has already been addressed in depth.
My only specific query is below:
Lines 215-218- I'm not sure the objection against pain relief in research animals is due to pain reduction invalidating the experiment per se. I believe it is more usually that the pain management used may impact on some experimental outcomes eg inflammatory markers or healing therefore making it difficult to interpret the experiment.
Section this comment refers to:
„The basic idea is that for many experiments, reducing or eliminating research animals’ pain would threaten the validity of the experiment, and, hence, should be avoided. It is an argument that has been adduced against the use of pain relief via pharmacological means in research animals. (2) It could also be adduced against our proposal.“
We agree with the reviewer and have reformulated the relevant section (see p. 5 lines 240-245) to make it more accurate:
If pain relief is known to have a negative, unquantifiable, uncontrollable effect on some study parameters than it becomes difficult to impossible to interpret the experiment and get any informative value or valid knowledge out of it:
The results of an experiment are only valid for its specific parameters:
A: [experiment + pain relief = Knowledge x].
B: [experiment + no pain relief = Knowledge y].
The conclusion in A is only valid when taking into account that pain relief might have had an unknown effect on “knowledge x”, which basically means that pain relief questions the validity or informative value of the experiment.
The conclusion in B is thus, prima facie, more valid.
Based on this, we think that our statement is in agreement with the statement of this reviewer. However, to make our statement clearer we changed “threaten the validity” to “negatively affect the validity” of an experiment (see p. 5 lines 240-245)
I also disagree with the statement discussing a lack of use of pain management (l218). I'm not sure whether there is up-to-date research on this, but I suspect pain relief is probably the norm rather than the exception where pain is significant.
We think that the reviewer is correct in stating that in many experiments pain relief is used. We did not intend to imply otherwise. To prevent this misunderstanding we deleted the word „widespread“ in the relevant sentence (page 5, lines 240-245)
Additionally, the following sentences mention that there is a moral and legal framework for minimising harm and suffering and it is explained under what circumstances pain relief is omitted. (please see page 6 lines 246-256)
„Though the 3Rs and the requirement of a positive harm-benefit analysis in the Directive provide a legal and moral framework for the avoidance or minimization of harm and suffering inflicted on research animals, they do not imply that causing or allowing pain in research animals is categorically prohibited. It can be morally and legally justified to do so if it is required to protect the validity of the experiment (4, article 14). For example, in the study of pain killers, pain itself is the studied parameter. The study objective therefore requires that the control cohort is left untreated while the other cohort is treated with the pain killer. However, pain inflicted on research animals may also be a side effect and researchers are then faced with a choice whether to provide pain relief or not. Take, for example, researchers growing cancers in animals to study cancer treatments. The pain these cancers cause can be difficult to treat with painkillers; successful pain treatment may require an intravenous catheter for round-the-clock medication with strong opioids“

Round 2
Reviewer 1 Report
No further comments to authors
Author Response
Reviewer comments are in black
Authors’ response in red
I think it should be accepted with amendments suggested below. None distract from the overall hypothesis and which is worth pursuing.
I was surprised that no reference was made to Latimer who was one of the first to update this
idea from the Uehiro
Institute? http://blog.practicalethics.ox.ac.uk/2018/03/oxford-uehiro-prize-in-practical-ethi
cs-why-we-should-genetically-disenhance-animals-used-in-factory-farms/
Shriver’s first paper dates back to 2009, while Latimer’s essay was only published last year. However, we have now added a reference to Latimer’s essay in the revised manuscript (see p. 2).
Line 42 environment” (4, article 38 d), should be Article 38.2.(d).
The manuscript has been changed accordingly-
Line 89: The authors should mention some of the unpredictable unwanted side effects of CrispR that are being discovered.
https://www.sciencealert.com/crispr-editing-causes-frequent-extensive-mutations-gene tic-damage-target-deletion-site
We would like to point out that the main papers the article in the link above is referring to were retracted because the authors failed to reproduce their own results.
However, we acknowledge the potential risk of off-target effects and have changed our manuscript accordingly (see footnote III, p. 3).
Lines 137 et seq: not all injuries result in pain but for example an infection of a wound could be lethal, or lead to a resistant bacterium causing a problem for other animals in close contact i.e. become a group/herd/flock etc problem.
The reviewer is referring to the following sentences:
„However, some may deny that the ablation of the affective pain dimension would be compatible with Refinement by arguing that this is likely to result in an increase in injuries, such as bruising, and this may adversely affect the animal’s wellbeing. However, as mentioned earlier, studies have shown that eliminating the affective pain dimension can leave acute pain features and acute responses to noxious stimulation intact (13). Thus, animals are likely to still exhibit normal guarding behaviour (i.e., behaviour aimed at avoiding pain), making an increase in self-injuries unlikely.“
3
We agree with the reviewer that an infection of a wound could be lethal and become a flock problem.
However, the point of the argument is that an increase in self-injuries is unlikely; hence there should not be an increased risk for wounds and thus, no increased risks for infections.
Or to rephrase, genetically disenhancened animals will have a similar risk for infections compared to their unaltered ‘kin’.
Line 353 5. Russel, W. & Burch, R. should be Russell
The manuscript has been changed accordingly.
Line 192: this is factually inaccurate, for example in the EU there are several Directives aiming to phase out poor welfare husbandry practices e.g. veal crates, caging of hens, antibiotic use., testing of cosmetics.
The reviewer is referring to the following argument:
“we think it was justified, and even morally obligatory to improve the slaves’ lives whenever the improvement was significant and there was no clear evidence that it would significantly delay abolition. Utilitarians may disagree, but even they may think that we should have done all we can to increase the welfare of slaves at the time when the risk of this causing a delay in the abolition of slavery was small, or uncertain. Since it is uncertain whether improving the welfare of research animals through genetic disenhancement will effectively delay the transition to the abolition of animal experiments, we think we should do it.”
Unfortunately, we do not quite understand why the reviewer is referring to Directives aiming to phase out poor welfare husbandry? We did not want to imply that there are no measures taken to increase animal welfare in agriculture or animal research.
The point of our argument here is that opponents have raised objections to practices that increase animal welfare on the grounds that it delays the abolition of a morally unacceptable practice. We argue that a) even if disenhancement delays phasing out animal experimentation, we still have a moral duty to improve animal welfare and b) the claim that disenhancement will in fact delay the phasing out of animal experimentation is a difficult to prove empirical claim.
4
249-251: The pain these cancers cause can be difficult to treat with painkillers; successful pain treatment may require an intravenous catheter for round-the-clock medication with strong opioids. A reference is required as I know of no cancer research that has done this? Maybe leave it as a hypothetical or use arthritis as an example? In any event usually implementing humane endpoints at an earlier stage would intervene.
We have added a reference.
The argument used by Referee 1 over Intrinsic Value would take a lot of discussion, and then would not alter the main thrust of this paper. Intrinsic value could potentially argue against using animals at all, but again that is not the point of this paper.
5
